# Effect of RNAi Targeting *SeGrx1* on the Cytotoxicity and Insecticide Susceptibility of Camptothecin in *Spodoptera exigua*

Fulai Yang, Lan Zhang *, Yanning Zhang, Liangang Mao, Lizhen Zhu, Xingang Liu and Hongyun Jiang

State Key Laboratory for Biology of Plant Diseases and Insect Pests, Institute of Plant Protection, Chinese Academy of Agricultural Sciences, Beijing 100193, China; wlyang1148@126.com (F.Y.); ynzhang@ippcaas.cn (Y.Z.); lgmao@ippcaas.cn (L.M.); lzzhu@ippcaas.cn (L.Z.); xgliu@ippcaas.cn (X.L.); hyjiang@ippcaas.cn (H.J.)
* Correspondence: lanzhang@ippcaas.cn

**Abstract:** Glutaredoxins (Grxs) are a class of small, heat-stable, acidic proteins which have been implied in various biological activities in cells, including the defense against oxidative stress induced by various biotic and abiotic factors. In this paper, the effects of RNAi targeting *SeGrx1* on the cytotoxicity and insecticide susceptibility of camptothecin (CPT) in *Spodoptera exigua* were investigated. Results showed that the cytotoxicity of CPT to the cells of *S. exigua* is heightened significantly by the silencing of *SeGrx1*. In the larvae of *S. exigua*, the mortality was significantly increased compared to CPT-alone treatment group at 120 h after knocking down the *SeGrx1* gene. Taken together, our results confirmed that *SeGrx1* in *S. exigua* played an important role in protecting the cells from the cytotoxicity induced by CPT, and the sensitivity of *S. exigua* larvae to CPT was increased by the silencing of *SeGrx1*. Our findings might provide basic information for understanding the function of Grxs and a strategy in insect pest control of RNAi technology combined with pesticides.

**Keywords:** glutaredoxin; cell viability; dsRNA injection; bioassays; insect control



## 1. Introduction

Glutaredoxins (Grxs) are a class of small and heat-stable oxidoreductases conserved in viruses, eukaryotes, and prokaryotes [1]. They belong to the thioredoxin superfamily and have been proved to play essential functions in cellular redox homeostasis [2]. Until now, eight members of Grxs were found in *Saccharomyces cerevisiae* and five in *Homo sapiens*, and they are divided into three major categories based on the structure and catalytic properties [3,4]. The first group, referred to as class I, consists of Grx1 and Grx2, which are the typical Grxs with a characteristic Cys-X-X-Cys (CXXC) active site motif and a thioredoxin/glutaredoxin fold [3]. Grx1 exists widely in the mitochondrial inner membrane space, nucleus, and cytoplasm, and plays a role in oxidative stress and redox homeostasis [1,5]. It has been implicated in the regulation of many cellular processes including apoptosis, oxidation, and inflammation, and is related closely to aging and the pathogenesis of diabetes and cardiovascular diseases caused by oxidative stress in humans [6]. Grx1 can play the regulatory roles through both its enzymatic redox activity and protein–protein interactions with specific proteins, such as Ask1, Ras, Fas, and procaspase-3, which are involved in the apoptosis signal transduction pathway [7]. It has been proved by a growing body of evidence that there is potential clinical and therapeutic application of Grx1 in atherosclerosis, and neurodegenerative disease, and aging-related diseases [8].

In insects, several studies were focused on the identification and classification of Grxs, and few studies investigated the role of Grxs in oxidative stress induced by exposure to biotic and abiotic factors. In order to identify the major components of the antioxidant system in *Apis mellifera*, three members of Grxs, *Grx1*, *Grx2* and *Grx-like-1* were identified by using genome sequence and comparative analysis [9]. In *A. cerana cerana*, two glutaredoxin genes, *AccGrx1* and *AccGrx2*, were identified and then investigated for their response

to abiotic environmental stress such as temperature, $H_2O_2$, and pesticides [10]. This study demonstrated that *AccGrx1* and *AccGrx2* play important roles in antioxidant defense when *A. cerana cerana* is subjected to oxidative stress [10]. Until now, four genes *HaGrx*, *HaGrx3*, *HaGrx5*, and *HaGdccr* were identified in *Helicoverpa armigera* and their role in protecting insects against oxidative stress induced by temperature and $H_2O_2$ treatments were confirmed successively [11,12]. In *Ostrinia furnacalis*, adverse environments (including starvation, ultraviolet light, mechanical injury, *Escherichia coli* exposure, and high and low temperatures) dramatically induced the transcript expression of OfurGRXGRX2 [13]. These results confirmed that Grxs play important roles in antioxidant defense in insects, as well as highlighted the need and importance for further indepth research in the physiological function of the insect Grxs.

Camptothecin (CPT), an indole alkaloid isolated from *Camptotheca acuminate* Decaisnean, showed significant biological activities against several insect pests including *Brevicoryne brassicae*, *Empoasca vitis*, *Nilaparvata lugens*, *Chilo suppressalis*, and *Heliothis virescens*, which suggested its potential use as a pesticide in the field [14,15]. CPT strongly inhibits the growth, development, and reproduction of *S. exigua* Hübner larvae [14]. Moreover, it can induce cytotoxic effects against IOZCAS-Spex-II cells derived from *S. exigua* by promoting the increase of intracellular oxidative stress due to the accumulation of intracellular ROS [16,17]. In our previous study, the full-length cDNA of *SeGrx1* was cloned (GenBank accession no.: MK318813) and expressed successfully in vitro, and its enzymatic kinetic parameters were obtained, which provided a foundation for further exploring the biological function of *SeGrx1* [18]. In this study, we conducted RNA interference (RNAi) experiments to investigate the role of *SeGrx1* on the cytotoxicity and insecticide susceptibility of CPT in *S. exigua*.

## 2. Materials and Methods

### 2.1. Cell Lines and Insects

The IOZCAS-Spex-II cells used in this present study were cultured in Grace's insect culture medium (Invitrogen Life Technologies, New York, NY, USA) supplemented with 10% fetal bovine serum (Invitrogen Life Technologies, New York, NY, USA) in T25 cm$^2$ tissue culture flasks (Corning, New York, NY, USA) at 27 °C, and were acquired from the Institute of Zoology, Chinese Academy of Sciences (Beijing, China) [13]. The third-instar larvae of *S. exigua* laboratory strain were obtained from Dr. Cui at the Institute of Plant Protection Chinese Academy of Agricultural Sciences.

### 2.2. Total RNA Extraction and cDNA Synthesis

Total RNA was extracted from the IOZCAS-Spex-II cells on logarithmic phase using RNasy Mini Kit (QIAGEN, Duesseldorf, Germany) following the manufacturer's protocol, and the quantity and quality of RNA were assessed by using Infinite M200 Pro NanoQuant (Tecan Trading, Männedorf, Switzerland) and agarose gel electrophoresis. According to the manufacturer's protocol, 1 µg total RNA was used to synthesize the first-strand cDNA with an EasyScript cDNA Synthesis Supermix Kit (Transgen, Beijing, China).

### 2.3. Double-Stranded RNA (dsRNA) Synthesis

Primers for dsRNA synthesis corresponding to *SeGrx1* and *SeGFP* were designed using the Primer-BLAST online tool (http://www.ncbi.nlm.nih.gov/tools/primer-blast/ (accessed on 22 November 2018), Table 1) and then synthesized by BGI genomics Co., Ltd. (Beijing, China). The synthesis and purification of ds*SeGrx1* and ds*SeGFP* were performed according to the instructions of the T7 RiboMAX$^{TM}$ Express RNAi System kit (Promega, Madison, WI, USA), after which the quantity and quality of dsRNA were analyzed by Infinite M200 PRO (Tecan, Männedorf, Switzerland) and 1.5% agarose gel electrophoresis, respectively [19,20].

**Table 1.** Primers used for dsRNA synthesis and RT-qPCR.

| Purpose | Name | Sequences (5′-3′) | Size (bp) |
|---|---|---|---|
| dsRNA syntheses | ds*SeGrx1* F | <u>TAATACGACTCACTATAGGGA</u>GAGGGCTCTCTGGCAAGCAAAA | 198 |
| | ds*SeGrx1* R | <u>TAATACGACTCACTATAGGGA</u>GAGTCGCGCTCATCCAGTTCAT | |
| | ds*GFP* F | <u>TAATACGACTCACTATAGGGA</u>AGTTCAGCGTGTCC | 520 |
| | ds*GFP* R | <u>TAATACGACTCACTATAGGGA</u>CTTCTCGTTGGGGTC | |
| RT-qPCR | SeGrx1 F | GCGATTCAAGAAAACCTGGCT | 130 |
| | SeGrx1 R | AGCATGGGCTCTAGTTTGCC | |
| | α-Tubulin F | GGAAGGAGAGTTCTCCGAGG | 152 |
| | α-Tubulin R | GGGGAATGTATTACGGTGCG | |
| | GAPDH F | GAAAACACCGGTGGACTCAA | 134 |
| | GAPDH R | GGCACCGTTGATATGCAAGA | |

Note: The underlined sequence added at the 5′end of the primer is the T7 promoter sequence.

### 2.4. Effect of RNAi on the Cytotoxicity of CPT

#### 2.4.1. Cell Transfection with ds*SeGrx1*

For cell transfection [21], the normal cells on logarithmic phase were harvested at a density of $10^5$ cells/mL, which were incubated overnight in 6-well transparent plates (1800 μL/well) without fetal bovine serum and antibiotics. The transfection of 10 μg ds*SeGrx1* and ds*GFP* was conducted using Lipofectamine® 2000 (Thermo Fisher Scientific, Waltham, MA, USA) according to the manufacturer's instructions, respectively. Each RNAi treatment was replicated at least three times. The transfection efficiency was detected by RT-qPCR.

#### 2.4.2. CPT Exposure

An amount of 10 μM CPT was added into each group after the cells were treated with ds*GFP* or ds*SeGrx1*, respectively. Cells treated with 0.1% DMSO were conducted as the control group [13]. Subsequently, cells of different treatment were collected at 2, 4, 6, 12, 24, and 48 h for the following morphological observation immediately or stored at −80 °C. The morphological changes of IOZCAS-Spex-II cell were recorded by an inverted phase contrast microscope (IX53, Olympus, Japan).

#### 2.4.3. Cell Viability Assay

The proliferative activity of cells was detected with a Cell Titer 96 Aqueous One Solution Cell Proliferation Assay Kit (Promega, Madison, WI, USA). According to the manufacturer's instruction, IOZCAS-Spex-II cells were collected at the certain time in 96-well plates (170 μL/well) at a density of $1.00 \times 10^5$ cells/ mL. An amount of 30 μL CellTiter 96®AQueous One solution was added to each well and then incubated for 2 h at 27 °C. The formazan product was measured at 490 nm using an Infinite M200 PRO microplate reader (Tecan, Männedorf, Switzerland) [13].

### 2.5. CPT Sensitivity against S. exigua after RNAi

#### 2.5.1. dsRNA Injection

The method of injection was used to introduce dsRNA into the third-instar larvae of *S. exigua* with a nanoliter injector (WPI, Beijing, China). A total of 2.5 μL ds*SeGrx1*, ds*GFP* or diethylpyrocarbonate (DEPC)-treated water was injected into the third-instar larvae. Each treatment was replicated three times, and for each replication, 20 larvae were injected [22]. RNA was extracted after 24 h to check the efficiency of RNAi by RT-qPCR.

#### 2.5.2. Bioassays

The third-instar larvae of *S. exigua* were subjected to bioassays after 24 h postinjection by using leaf-dipping method [23]. Briefly, the cabbage leaf discs (7 cm diameter) were cut and dipped in 3.48 mg/L CPT distilled water solutions containing 0.1% DMSO for 30 s and then air dried for 1 h at room temperature. Leaf discs were placed in petri dish (9 cm

diameter) and a total of 20 third-instar larvae were introduced into each dish. The control group larvae were fed with leaf discs treated with 0.1% DMSO. Each group was replicated three times. All bioassays were conducted at 25 ± 1 °C, 50–60% RH and under a 16:8 h (light/dark) photoperiod. The mortalities were recorded at 96 and 120 h, and the leaf discs were replaced every day during the bioassays. Additionally, weights of the survivors were measured at 120 h.

### 2.6. Gene Expression with RT-qPCR

Primers used for the RT-qPCR analysis were designed with Primer Premier 6.0 and synthesized by BGI genomics Co., Ltd. (Beijing, China) (Table 1). Real-time PCR was conducted using a QuanStudio 3 ABI system (Thermo Fisher Scientific, Waltham, MA, USA) with the TransStart Top Green qPCR SuperMix kit (Transgen, Beijing, China). According to the manufacturer's instructions, a total volume of 20 μL reaction mixture containing 1 μL of cDNA template, 1 μL of each primer, 10 μL of 2 × TransStart® Top Green qPCR SuperMix, and 7.0 μL of distilled ddH$_2$O. The RT-qPCR conditions were 30 s at 94 °C, followed by 40 cycles at 94 °C for 30 s, and then annealing at 72 °C for 30 s. The relative expression was calculated with α-tubulin and GADPH as reference genes according to the methods developed by Vandesompele et al. [24].

### 2.7. Statistical Analysis

All results were confirmed in at least three independent experiments. Data are presented as mean ± standard error. The SPSS 26.0 Software Package (SPSS Inc., Chicago, IL, USA) was used to perform statistical analyses. Independent samples t-test and one-way ANOVA followed by the Duncan's multiple range test were performed. Means with the different letters are significantly different at $p < 0.05$.

### 3. Results and Discussion

#### 3.1. RNAi Targeting SeGrx1 Increased the Cytotoxicity of CPT in IOZCAS-Spex-II Cells

The effect of dsRNA on the mRNA level of *SeGrx1* expression change *in* IOZCAS-Spex-II cells after transfection with ds*SeGrx1* was detected using RT-qPCR. The expression level of *SeGrx1* in the treatment group decreased 6.33, 56.0, 98.3, and 74.7% compared to the control group transfected with Lipofectamine 2000 Reagent, respectively. Moreover, the relative expression levels of *SeGrx1* were changed from 0.99 to 1.20 compared to the control group. This result suggests that transfection of specific ds*SeGrx1* is an effective way to silence the expression of *SeGrx1* in IOZCAS-Spex-II cells (Figure 1). Therefore, the CPT was added into the cells transfected with ds*SeGrx1* or ds*GPF* for 24 h to ensure the RNAi efficiency. As shown in Figure 2, there was no significant difference in cell morphologies between cells treated with 0.1% DMSO and disrupted with ds*GFP*. The cell treated with ds*SeGrx1* showed some certain morphological changes with apoptotic bodies at 24 and 48 h [13]. In contrast, the cells treated with CPT and dsSe*Grx1* + CPT showed typical characteristics of apoptosis in their morphological changes, such as cell shrinkage, gap generation, membrane blebbing, and apoptotic bodies. The inhibition rate of cell viability induced by ds*GFP* fluctuated between 1.69% and 17.8% (Figure 3). The efficacy of ds*SeGrx1* on the cell viability increased gradually with time and reached to 29.6% at 48 h. Consistent with previous reports, CPT showed cytotoxic effects to IOZCAS-Spex-II cells in a time-dependent manner with the inhibition rate increasing from 4.64% at 2 h to 48.1% at 48 h (Figure 3). It was noteworthy that the inhibition of cell viability is heightened significantly by the disruption of *SeGrx1* together with the treatment of 10 μM CPT in IOZCAS-Spex-II cells at 6, 12, 24, and 48 h (Figure 3). These results suggest that *SeGrx1* may function to protect IOZCAS-Spex-II cells from CPT-induced apoptosis, which confirms previous reports that *SeGrx1* plays important roles in antioxidant defense in insects [16,25]. In our previous studies, a significant increase in the level of intracellular ROS was observed, accompanied by markedly increased DNA damage, lipid peroxidation, and protein carbonylation after

exposure to CPT in IOZCAS-Spex-II cells [16]. It could be proposed that oxidative stress is more intense in SeGrx1-silenced IOZCAS-Spex-II cells.

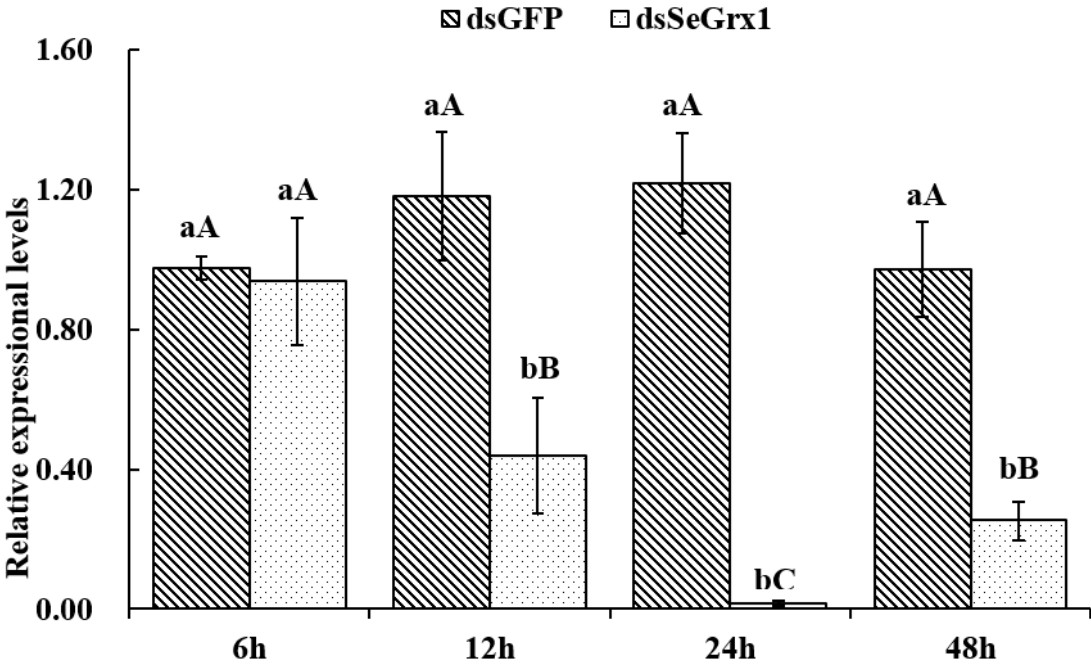

**Figure 1.** The relative expression of *SeGrx1* after RNAi in IOZCAS-Spex-II cells. Data are presented as mean ± standard error. Independent samples t-test and one-way ANOVA, followed by a Duncan's multiple range test were performed by using SPSS 26.0 Software Package (SPSS Inc., Chicago, IL, USA). Data followed by the same lowercase letters indicate no significant difference at 0.05 level between dsGFP group and dsSeGrx1 treatment at the same time. Data followed by the same capital letters indicate no significant difference at 0.05 level among the same group at different times.

Previous studies demonstrated that *Grx1* is highly sensitive to oxidants and specifically catalyzes the reduction of specific target proteins, which have important functions of cell protection and antioxidation [6,7,26]. In *A. cerana cerana*, it has been showed that *AccGrx1* may play critical roles in antioxidant defense against lower temperature (4 °C), $H_2O_2$, $HgCl_2$, and pesticide cyhalothrin and phoxime treatments [10]. Our previous studies showed that CPT treatment can induce the overproduction of ROS accompanied by markedly increased DNA damage, lipid peroxidation, and protein carbonylation in IOZCAS-Spex-II cells [16]. These results indicated that oxidative stress induced by CPT played an essential role in the toxicity and mode of action of CPT at the cellular level. In this study, the cytotoxicity of CPT was heightened significantly by the disruption of *SeGrx1* together with the treatment of 10 µM CPT in IOZCAS-Spex-II cells. These results suggest that *SeGrx1* may function to protect IOZCAS-Spex-II cells from CPT-induced oxidative stress, which confirms previous reports that Grx1 plays important roles in antioxidant defense in insects [26]. A similar result was reported that the levels of ROS are controlled by the activities of *Grx2* in mitochondria, which can help to modulate the susceptibility of a cell to apoptosis [27]. Prior studies have noted the importance of Grx2 in mitochondrial redox status, when Grx2 knock-down resulted in increasing the sensitivity to cell death induced by doxorubicin/adriamycin and phenylarsin [28]. Taken with the above studies, Grxs as the major antioxidant enzyme families were involved in regulating cellular redox homeostasis and in defense of enhanced oxidative stress induced by adverse factors including temperatures, ultraviolet light, pesticides, and so on. In this study, we confirmed that *SeGrx1* is involved in the defense of CPT-induced oxidative stress in IOZCAS-Spex-II cells.

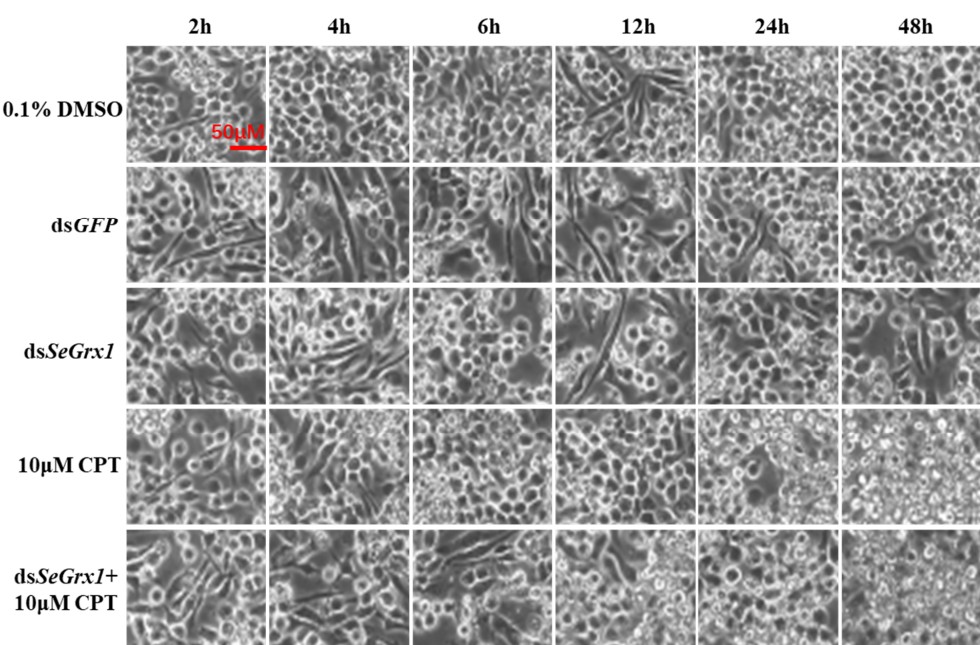

**Figure 2.** Morphology observation of IOZCAS-Spex-II cells treated with ds*GFP*, ds*SeGrx1*, 10 μM CPT, and de*SeGrx1* + 10 μM CPT. 0.1% DMSO was used as a control. The scale bar is 50 μm. The morphological changes of IOZCAS-Spex-II cell were recorded by an inverted phase contrast microscope with 400× magnification (IX53, Olympus, Tokyo, Japan).

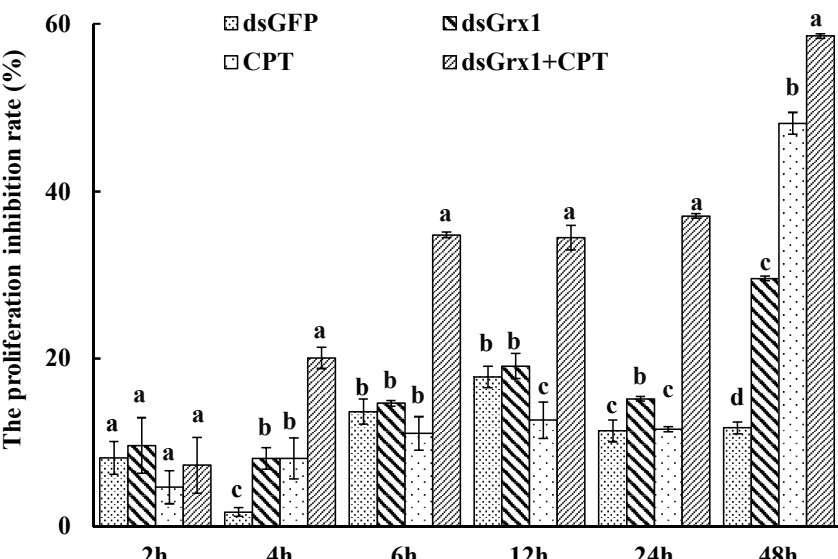

**Figure 3.** Inhibitory effects of dsRNA and CPT to *S. exigua* cell line IOZCAS-Spex-II. Inhibition (%) = (OD490 of the 0.1% DMSO-treated cells − OD490 of CPT or/and dsRNA-treated cells)/OD of 0.1% DMSO-treated cells × 100%. Values are shown as mean ± SEM. Data followed by the same lowercase letters among different treatment at the same time indicate no significant difference at 0.05 level.

### 3.2. RNAi Targeting SeGrx1 Increased the Sensitivity of CPT against S. exigua

The RT-qPCR results showed that the expression levels of *SeGrx1* were significantly downregulated at 24 h after silencing with 1.50, 3.00, and 6.00 μg/larvae ds*SeGrx1*, respectively (Figure 4). The highest RNAi efficiency was 72.1% at the concentration of 1.5 μg/larvae ds*SeGrx1*. This result suggested that RNAi of specific ds*SeGrx1* is also an effective way to silence the expression of *SeGrx1* in the larvae of *S. exigua*. Furthermore, the bioassays test was conducted to examine the effect of RNAi on the survival and weight of CPT-treated *S. exigua* larvae. The cumulative mortality of *S. exigua* larvae was 6.32, 32.6,

50.6, and 72.9% in the ds*GFP*, ds*SeGrx1*, CPT, and ds*SeGrx1* + CPT group, respectively (Figure 5) The results showed that after knocking down the *SeGrx1* for 24 h, the mortality was significantly increased compared to the CPT-alone treatment group at 96 h, which suggested that RNAi targeting *SeGrx1* increased the sensitivity of CPT against *S. exigua*. In addition, the weight of the survivors was measured (Figure 5). Interestingly, the weight of the surviving larvae treated with ds*SeGrx1* alone was increased significantly. However, after feeding with leaves treated with CPT, the weight of the *S. exigua* larvae decreased compared to that treated with CPT alone, although there was no significant difference ($p > 0.05$) (Figure 6). These observations are consistent with previous findings in other insects, including *Spodoptera frugiperda* [29,30]. CPT diets induced weight loss of the larvae of *S. frugiperda* and the molecular basis for the impact of CPT on *S. frugiperda* was explored by comparative transcriptomic analyses among midgut samples. Our results suggested that the inhibition effect on the growth of CPT can be increased by the knocking down of *SeGrx1*, which showed that CPT as an insecticide may be used with other insecticides for enhanced efficiency in controlling important insect pests in the field. In *Homo sapiens*, Grxs have been implicated in various physiological and pathological conditions, from immune defense to neurodegeneration and cancer development, which makes Grxs a possible drug target [8,31]. The RNAi-mediated gene knockdown has shown promising results in different insect groups, pointing it to be the upcoming technique for insect control [32–36]. According to the reports of Yoon [37], the inhibitor of apoptosis (IAP) protein, a negative regulator of apoptosis in insects, provides opportunities for developing targets for RNAi-based insect pest control. In this study, RNAi targeting *SeGrx1* increased the sensitivity of CPT against *S. exigua*. According to the reports of Liu et al. (2021), after GmGrx is silenced by RNAi, the percentage of larval survival to emamectin benzoate was significantly decreased, demonstrating that GmGrx contributes to the defense of oxidative damage induced by emamectin benzoate in *Grapholita molesta* (Busck) [38]. These results provided insights for an innovative strategy in insect control of RNAi technology with the silencing of Grxs combined with pesticides.

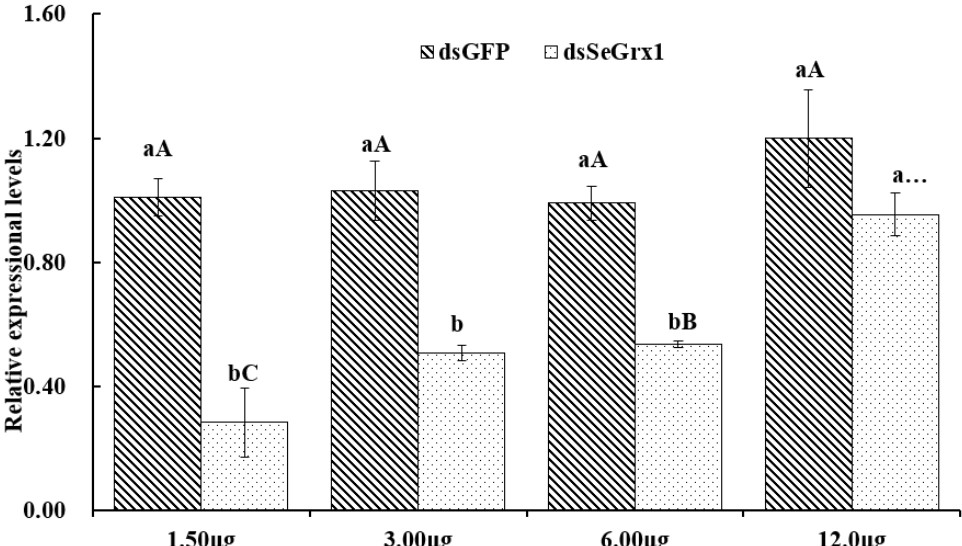

**Figure 4.** The relative expression of *SeGrx1* after RNAi in the larvae of *S. exigua*. Data are presented as mean ± standard error. Independent samples t-test and one-way ANOVA, followed by the Duncan's multiple range test were performed by using SPSS 26.0 Software Package (SPSS Inc., Chicago, IL, USA). Data followed by the same lowercase letters indicate no significant difference at 0.05 level between ds*GFP* group and ds*SeGrx1* treatment at the same time. Data followed by the same capital letters indicate no significant difference at 0.05 level among the same group treated with different concentration of dsRNA.

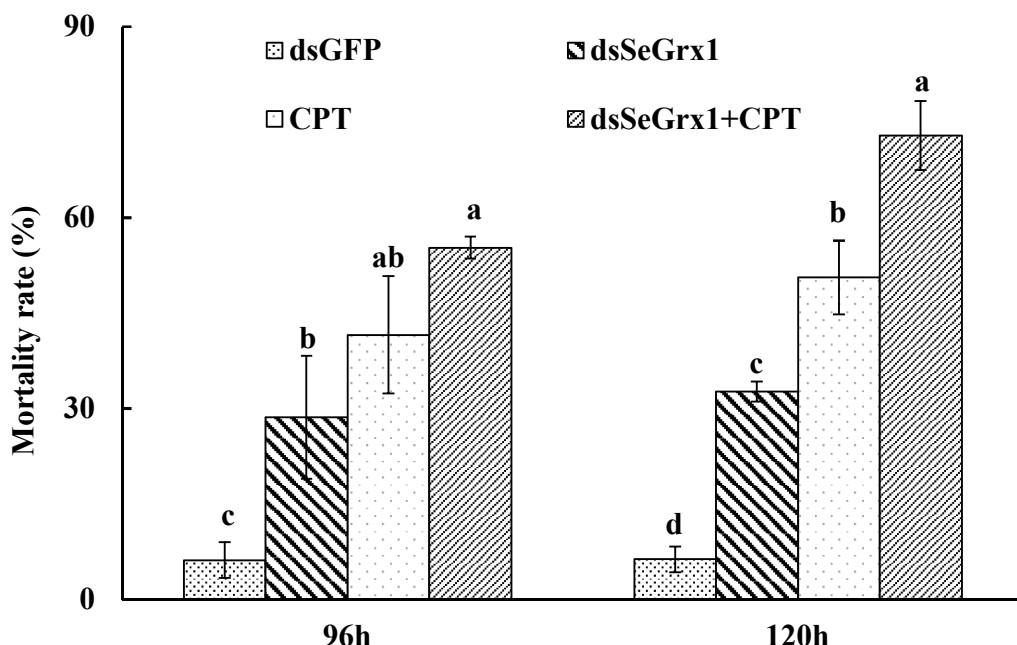

**Figure 5.** Mortality of *S. exigua* larvae treated by dsRNA and/or CPT at 96 h and 120 h. Values are shown as mean ± SEM. Data followed by the same lowercase letters among different treatments at the same time indicate no significant difference at 0.05 level.

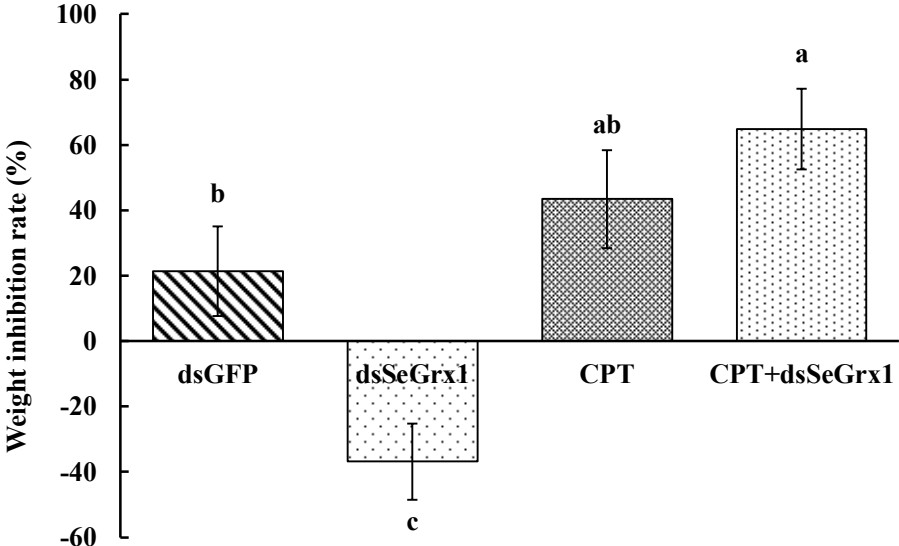

**Figure 6.** Weight inhibition rate of *S. exigua* larvae treated by dsRNA and/or CPT at 120 h. Values are shown as mean ±SEM. Data followed by the same lowercase letters indicate no significant difference at 0.05 level.

### 4. Conclusions

In conclusion, our results confirmed that *SeGrx1* played an important role in defending against the oxidative stress induced by CPT in *S. exigua*, and the sensitivity of larvae to CPT was increased by the silencing of *SeGrx1*. It could be proposed that oxidative stress is more intense in insects. These results provide a strategy in insect pest control of RNAi technology combined with pesticides.

**Author Contributions:** F.Y. and L.Z. (Lan Zhang) conceived and designed the research. F.Y. and Y.Z. performed the experiments. F.Y. analyzed the data and wrote the manuscript. L.Z. (Lizhen Zhu) and H.J. reviewed the manuscript. X.L. and L.M. helped analysis with constructive discussions. All authors have read and agreed to the published version of the manuscript.

**Funding:** This research was supported by the Chinese National Natural Science Foundation (31672059).

**Institutional Review Board Statement:** Not applicable.

**Informed Consent Statement:** Not applicable.

**Data Availability Statement:** Not applicable.

**Acknowledgments:** We would like to thank Li Cui for kindly providing *S. exigua* larvae.

**Conflicts of Interest:** The authors declare no conflict of interest.

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
