# Peer review of "Effect of RNAi Targeting SeGrx1 on the Cytotoxicity and Insecticide Susceptibility of Camptothecin in Spodoptera exigua"

_agriculture, doi:10.3390/agriculture12070930_

Round 1
Reviewer 1 Report
The manuscript concerns the effect of RNAi targeting SeGrx1 on the cytotoxicity and insecticide susceptibility of camptothecin in Spodoptera exigua.
Authors show that the cytotoxicity of CPT to the cells of S. exigua is heightened significantly by the silenced of SeGrx1.
Their results confirmed that SeGrx1 in S. exigua played an important role in protecting the cells from the cytotoxicity induced by CPT, and the sensitivity of S. exigua larvae to CPT was increased by the silence of SeGrx1.
The manuscript is clear, relevant for the field and presented in a well-structured manner.
This manuscript is relevant and of interest to the scientific community because their conclusions might provide a basic information for understanding the function of Grxs and a strategy in insect pest control of RNAi technology combined with pesticides.
I only noticed some small errors in the legend of the figure 2:
- The scale bar is not present.
- “deSeGrx” need to be change to “dsSeGrx”
Author Response
Response to Reviewer 1 Comments
Point 1:The manuscript concerns the effect of RNAi targeting SeGrx1 on the cytotoxicity and insecticide susceptibility of camptothecin in Spodoptera exigua. Authors show that the cytotoxicity of CPT to the cells of S. exigua is heightened significantly by the silenced of SeGrx1.Their results confirmed that SeGrx1 in S. exigua played an important role in protecting the cells from the cytotoxicity induced by CPT, and the sensitivity of S. exigua larvae to CPT was increased by the silence of SeGrx1. The manuscript is clear, relevant for the field and presented in a well-structured manner. This manuscript is relevant and of interest to the scientific community because their conclusions might provide a basic information for understanding the function of Grxs and a strategy in insect pest control of RNAi technology combined with pesticides.
Response 1: Thank you for your kindly comments.
Point 2: I only noticed some small errors in the legend of the figure 2:
- The scale bar is not present.
- “deSeGrx” need to be change to “dsSeGrx”
Response 2: The scale bar is added to Figure 2.
Reviewer 2 Report
The paper is interesting, and the experiments are well organized, but I raised some issues, especially in figures. Moreover, there are a lot of English errors, I just reported some of them.
KEYWORDS
Please substitute these keywords as they are already in the title Camptothecin; RNA interference; Spodoptera exigua
INTRODUCTION
Line 26, write “which are divided into three major categories”
Line 27, write “The first group, referred to as class I, consists”
Line 28, please specify what a “CXXC active site motif” is.
Line 37, write “which are involved in apoptosis signal transduction”
Line 40, write “few studies investigated”
Line 53 “Camptothecin (CPT), an indole alkaloid isolated from Camptotheca acuminate Decaisnean, showed significant biological activities against several insect pests including Brevicoryne brassicae, Empoasca vitis, Nilaparvata lugens and Chilo suppressalis, which suggested its potential use as a pesticide in the field [13].” The effect was also evaluated on Heliothis virescens, the tobacco budworm, hemocytes (https://doi.org/10.1038/s41598-017-11939-x)
Line 56, write “CPT strongly inhibits”
Line 57, correct “developement” with “development”
MATERIAL AND METHODS
Line 120, write “Leaf discs were placed in petri 120 dish (9 cm diameter) with a total of a 20 third-instar larvae introduced into each dish.”
Line 142, write “Means with different letters”
RESULTS AND DISCUSSION
Line 154 “As shown in Fig. 2, there was no significant difference in cell morphologies between cells treated with 0.1% DMSO and disrupted with dsGFP” please add in material and method section information on morphology observations.
Line 171 Please add references “In A. cerana cerana, it has been showed that AccGrx1 may play critical roles in antioxidant defense against lower temperature (4℃), H2O2, HgCl2 and pesticide cyhalothrin and phoxime treatments.”
Line 173 please add references “Our previous studies have showed that CPT treatment can induce the overproduction of ROS accompanied by markedly increased DNA damage, lipid peroxidation and protein carbonylation in IOZCAS-Spex-II cells.” Moreover write “have shown”.
Line 180 please add references “which confirms previous reports that Grx1 play important roles in antioxidant defense in insects.”
Line 195, write “of specific dsSeGrx1 is also an effective way to”
Line 203 “Interestingly, the weight of the survival larvae treated with dsSeGrx1 only was significantly.” I suggest to write “treated with dsSeGrx1 alone” and “was significantly” does not mean anything, please rephrase the sentence.
Line 205, write “larvae decreased compared”
Line 207, write “have been implicated”
Line 209 “The RNAi-mediated gene knockdown has shown promising results in different insect groups, pointing it to be the upcoming technique for insect control” please add more references (https://doi.org/10.1016/j.cub.2016.10.013, doi: 10.1038/s41598-018-20416-y)
Line 212, correct “apotoisis” with “apoptosis”
FIGURE CAPTIONS
Figure 1 please add information on the statistical test performed
Figure 2 please add information on which instrument was used and at what magnification
Figure 4, correct “exiuga” with “exigua”
Figure 5, write “different treatments”
FIGURE
Figure 1, the statistical analysis is not clear: was it performed between dsGFP group and dsSeGrx1 treatment at the same time or among the same group at different times? I suggest to perform both the analyses. Moreover, using uppercase and lowercase letters in this way is useless and confusing.
Figure 3 is not mentioned in the main text
Figure 4, same observations as figure 1.
Author Response
Response to Reviewer 2 Comments
Point 1:The paper is interesting, and the experiments are well organized, but I raised some issues, especially in figures. Moreover, there are a lot of English errors, I just reported some of them.
Response 1: Thank you for your kindly comments.
Point 2:Please substitute these keywords as they are already in the title Camptothecin; RNA interference; Spodoptera exigua
Response 2: Accepted. Keywords “Camptothecin; RNA interference and Spodoptera exigua” were substituted with “cell viability; dsRNA injection; bioassays; insect control”.
Point 3:Line 26, write “which are divided into three major categories”
Response 3: Accepted.
Point 4:Line 27, write “The first group, referred to as class I, consists”
Response 4: Accepted.
Point 5:Line 28, please specify what a “CXXC active site motif” is.
Response 5: Accepted. “CXXC active site motif” was changed to “Cys-X-X-Cys (CXXC) active site motif “.
Point 6:Line 37, write “which are involved in apoptosis signal transduction”
Response 6: Accepted.
Point 7:Line 40, write “few studies investigated”
Response 7: Accepted.
Point 8: Line 53 “Camptothecin (CPT), an indole alkaloid isolated from Camptotheca acuminate Decaisnean, showed significant biological activities against several insect pests including Brevicoryne brassicae, Empoasca vitis, Nilaparvata lugens and Chilo suppressalis, which suggested its potential use as a pesticide in the field [13].” The effect was also evaluated on Heliothis virescens, the tobacco budworm, hemocytes (https://doi.org/10.1038/s41598-017-11939-x).
Response 8: Accepted. This paper was cited.
Point 9: Line 56, write “CPT strongly inhibits”
Response 9: Accepted.
Point 10: Line 57, correct “developement” with “development”
Response 10: Accepted.
Point 11: Line 120, write “Leaf discs were placed in petri 120 dish (9 cm diameter) with a total of a 20 third-instar larvae introduced into each dish.”
Response 11: Accepted.
Point 12: Line 142, write “Means with different letters”
Response 12: Accepted.
Point 13: Line 154 “As shown in Fig. 2, there was no significant difference in cell morphologies between cells treated with 0.1% DMSO and disrupted with dsGFP” please add in material and method section information on morphology observations.
Response 13: Accepted. Lines 101-1-4 were changed to “Subsequently, cells of different treatment were collected at 2, 4, 6, 12, 24 and 48h for the following testsmorphological observation immediately or stored at -80℃. The morphological changes of IOZCAS-Spex-II cell were recorded by inverted phase contrast microscope (IX53, Olympus, Japan).”
Point 14: Line 171 Please add references “In A. cerana cerana, it has been showed that AccGrx1 may play critical roles in antioxidant defense against lower temperature (4℃), H2O2, HgCl2 and pesticide cyhalothrin and phoxime treatments.”
Response 14: Accepted.
Point 15: Line 173 please add references “Our previous studies have showed that CPT treatment can induce the overproduction of ROS accompanied by markedly increased DNA damage, lipid peroxidation and protein carbonylation in IOZCAS-Spex-II cells.” Moreover write “have shown”.
Response 15: Accepted.
Point 16: Line 180 please add references “which confirms previous reports that Grx1 play important roles in antioxidant defense in insects.”
Response 16: Accepted.
Point 17: Line 195, write “of specific dsSeGrx1 is also an effective way to”
Response 17: Accepted.
Point 18: Line 203 “Interestingly, the weight of the survival larvae treated with dsSeGrx1 only was significantly.” I suggest to write “treated with dsSeGrx1 alone” and “was significantly” does not mean anything, please rephrase the sentence.
Response 18: Accepted. The sentence was changed to “Interestingly, the weight of the survival larvae treated with dsSeGrx1 alone was increased significantly”.
Point 19: Line 205, write “larvae decreased compared”
Response 19: Accepted.
Point 20: Line 207, write “have been implicated”
Response 20: Accepted.
Point 21: Line 209 “The RNAi-mediated gene knockdown has shown promising results in different insect groups, pointing it to be the upcoming technique for insect control” please add more references (https://doi.org/10.1016/j.cub.2016.10.013, doi: 10.1038/s41598-018-20416-y)
Response 21: Accepted. More references were added.
Point 22: Line 212, correct “apotoisis” with “apoptosis”
Response 22: Accepted.
Point 23: Figure 1 please add information on the statistical test performed
Response 23: Accepted. The figure legend was changed to “The relative expression of SeGrx1 after RNAi in IOZCAS-Spex-Ⅱ cells. Data are presented as mean ± standard error. One-way ANOVA followed Duncans multiple range test were performed by using SPSS 26.0 Software Package (SPSS Inc., Chicago, IL, USA). Data followed by the same letters (little letters for dsGFP treatment group, capital letters for dsSeGrx1 treatment group) indicated no significant difference at 0.05 level”.
Point 24: Figure 2 please add information on which instrument was used and at what magnification.
Response 24: Accepted. Sentence “The morphological changes of IOZCAS-Spex-II cell were recorded by inverted phase contrast microscope with 400X magnification (IX53, Olympus, Japan) ” was added.
Point 25: Figure 4, correct “exiuga” with “exigua”
Response 25: Accepted.
Point 26: Figure 5, write “different treatments”
Response 26: Accepted.
Point 27: Figure 1, the statistical analysis is not clear: was it performed between dsGFP group and dsSeGrx1 treatment at the same time or among the same group at different times? I suggest to perform both the analyses. Moreover, using uppercase and lowercase letters in this way is useless and confusing.
Response 27: Accepted. Figure 1. The relative expression of SeGrx1 after RNAi in IOZCAS-Spex-Ⅱ cells. Data are presented as mean ± standard error. Independent Samples t-test and One-way ANOVA followed Duncans multiple range test were performed by using SPSS 26.0 Software Package (SPSS Inc., Chicago, IL, USA). Data followed by the same little letters indicated letters no significant difference at 0.05 level between dsGFP group and dsSeGrx1 treatment at the same time. Data followed by the same capital letters indicated no significant difference at 0.05 level among the same group at different times.
Point 28: Figure 3 is not mentioned in the main text
Response 28: Accepted. “Fig. 3” was added in the main text (Section 3.1).
Point 29: Figure 4, same observations as figure 1.
Response 29: Accepted. Figure 4. The relative expression of SeGrx1 after RNAi in the larvae of S. exigua. Data are presented as mean ± standard error. Independent Samples t-test and One-way ANOVA followed Duncans multiple range test were performed by using SPSS 26.0 Software Package (SPSS Inc., Chicago, IL, USA). Data followed by the same little letters indicated letters no significant difference at 0.05 level between dsGFP group and dsSeGrx1 treatment at the same time. Data followed by the same capital letters indicated no significant difference at 0.05 level among the same group treated with different concentration of dsRNA.

Reviewer 3 Report
The authors have studied the effect of RNAi-based silencing of the SeGrx1 gene in Spodoptera exigua on its susceptibility to camptothecin and showed that it causes significant impairments of insects’ tolerance systems leading to higher mortality rate and weight inhibition and inhibition of cell proliferation as well. I suppose that the theme of the research is quite relevant both in practical terms and in regard to fundamental science due to the fact that the role of antioxidative systems in insects which is well-studied, e.g. in humans and plants, and understudied in insects.
Nevertheless, I find that the article could not be accepted in its current form. The only major issue I could mention is the fact that the authors discuss the effect of SeGrx1 silencing in the context of impaired antioxidative systems and oxidative stress. In the previous study, the authors have clearly shown that camptothecin causes severe oxidative stress in Spodoptera cells. However, in the current study, only the mortality rate, cell morphology, and weight inhibition were studied. Of course, it is logically correct to suppose that in SeGrx1-silenced insects, camptothecin causes oxidative stress, while before the experimental check, it remains to be a hypothesis. Thus, I strongly recommend studying markers of oxidative stress in control and SeGrx1-silenced insects. I do not insist on performing a comprehensive analysis but one distinctive marker (e.g., lipid peroxidation rate of something else that is quite easy measurable) could be studied. With this experiment done, the study would be sufficiently more important for the field.
There are also some minor points related to the text.
1. Whereas the role of glutaredoxins in insects is poorly studied, I find that some articles were missed and could be discussed in the Introduction and/or Results sections:
Liu Y, Zhu F, Shen Z, Moural TW, Liu L, Li Z, Liu X, Xu H. Glutaredoxins and thioredoxin peroxidase involved in defense of emamectin benzoate induced oxidative stress in Grapholita molesta. Pestic Biochem Physiol. 2021 Jul;176:104881. doi: 10.1016/j.pestbp.2021.104881. Epub 2021 May 24. PMID: 34119223.
Mercer SW, Burke R. Evidence for a role for the putative Drosophila hGRX1 orthologue in copper homeostasis. Biometals. 2016 Aug;29(4):705-13. doi: 10.1007/s10534-016-9946-0. Epub 2016 Jul 5. PMID: 27379771.
Popović ŽD, Subotić A, Nikolić TV, Radojičić R, Blagojević DP, Grubor-Lajšić G, Koštál V. Expression of stress-related genes in diapause of European corn borer (Ostrinia nubilalis Hbn.). Comp Biochem Physiol B Biochem Mol Biol. 2015 Aug;186:1-7. doi: 10.1016/j.cbpb.2015.04.004. Epub 2015 Apr 14. PMID: 25882225.
An S, Zhang Y, Wang T, Luo M, Li C. Molecular characterization of glutaredoxin 2 from Ostrinia furnacalis. Integr Zool. 2013 Apr;8 Suppl 1:30-8. doi: 10.1111/j.1749-4877.2012.00301.x. Epub 2012 Oct 31. PMID: 23621469.
Giordano E, Peluso I, Rendina R, Digilio A, Furia M. The clot gene of Drosophila melanogaster encodes a conserved member of the thioredoxin-like protein superfamily. Mol Genet Genomics. 2003 Feb;268(5):692-7. doi: 10.1007/s00438-002-0792-0. Epub 2003 Jan 18. PMID: 12589444.
2. I suggest that the Discussion and especially the Conclusion sections should be sufficiently expanded. Authors could describe and discuss the effect of camptothecin on Spodoptera cells. The information could be found in the following articles:
Shu B, Yang X, Dai J, Yu H, Yu J, Li X, Cao L, Lin J. Effects of camptothecin on histological structures and gene expression profiles of fat bodies in Spodoptera frugiperda. Ecotoxicol Environ Saf. 2021 Nov 8;228:112968. doi: 10.1016/j.ecoenv.2021.112968. Epub ahead of print. PMID: 34763196
Shu B, Zou Y, Yu H, Zhang W, Li X, Cao L, Lin J. Growth inhibition of Spodoptera frugiperda larvae by camptothecin correlates with alteration of the structures and gene expression profiles of the midgut. BMC Genomics. 2021 May 26;22(1):391. doi: 10.1186/s12864-021-07726-8. PMID: 34039281; PMCID: PMC8157707.
Yang F, Wang L, Zhang L, Zhang Y, Mao L, Jiang H. Synthesis and biological activities of two camptothecin derivatives against Spodoptera exigua. Sci Rep. 2019 Dec 2;9(1):18067. doi: 10.1038/s41598-019-54596-y. PMID: 31792297; PMCID: PMC6889156.
Moreover, other the role of other antioxidative systems such as antioxidants, peroxidases, and thioredoxins could be discussed.
3. Please cite Primer-BLAST software
4. Please add the citation in line 175
5. The text requires considerable copy-editing due to misprints, spelling, and grammar errors (in human – in humans (line 33), homoeostasis -> homeostasis (line 24), a basic information -> basic information, etc.) there are too many issues for the reviewer to mention, thus I suggest re-checking the manuscript attentively to fix the errors accordingly.
Author Response
Response to Reviewer 3 Comments
Point 1:The authors have studied the effect of RNAi-based silencing of the SeGrx1 gene in Spodoptera exigua on its susceptibility to camptothecin and showed that it causes significant impairments of insects’ tolerance systems leading to higher mortality rate and weight inhibition and inhibition of cell proliferation as well. I suppose that the theme of the research is quite relevant both in practical terms and in regard to fundamental science due to the fact that the role of antioxidative systems in insects which is well-studied, e.g. in humans and plants, and understudied in insects. Nevertheless, I find that the article could not be accepted in its current form. The only major issue I could mention is the fact that the authors discuss the effect of SeGrx1 silencing in the context of impaired antioxidative systems and oxidative stress. In the previous study, the authors have clearly shown that camptothecin causes severe oxidative stress in Spodoptera cells. However, in the current study, only the mortality rate, cell morphology, and weight inhibition were studied. Of course, it is logically correct to suppose that in SeGrx1-silenced insects, camptothecin causes oxidative stress, while before the experimental check, it remains to be a hypothesis. Thus, I strongly recommend studying markers of oxidative stress in control and SeGrx1-silenced insects. I do not insist on performing a comprehensive analysis but one distinctive marker (e.g., lipid peroxidation rate of something else that is quite easy measurable) could be studied. With this experiment done, the study would be sufficiently more important for the field.
Response 1: Thank you for your kindly comments. In this study, the major aim is to investigate the role of the RNA interference of SeGrx1 on the cytotoxicity and insecticide susceptibility of CPT in S. exigua, which provide a basic information for understanding the function of Grxs and a strategy in insect pest control of RNAi technology combined with pesticides. In our previous studies, it has been shown a significant increase in the level of intracellular ROS accompanied by markedly increased DNA damage, lipid peroxidation and protein carbonylation after exposing to CPT in IOZCAS-Spex-II cells (Reference 16). In this paper, this reference has been cited.
Point 2:Whereas the role of glutaredoxins in insects is poorly studied, I find that some articles were missed and could be discussed in the Introduction and/or Results sections:
Liu Y, Zhu F, Shen Z, Moural TW, Liu L, Li Z, Liu X, Xu H. Glutaredoxins and thioredoxin peroxidase involved in defense of emamectin benzoate induced oxidative stress in Grapholita molesta. Pestic Biochem Physiol. 2021 Jul;176:104881. doi: 10.1016/j.pestbp.2021.104881. Epub 2021 May 24. PMID: 34119223.
Mercer SW, Burke R. Evidence for a role for the putative Drosophila hGRX1 orthologue in copper homeostasis. Biometals. 2016 Aug;29(4):705-13. doi: 10.1007/s10534-016-9946-0. Epub 2016 Jul 5. PMID: 27379771.
Popović ŽD, Subotić A, Nikolić TV, Radojičić R, Blagojević DP, Grubor-Lajšić G, Koštál V. Expression of stress-related genes in diapause of European corn borer (Ostrinia nubilalis Hbn.). Comp Biochem Physiol B Biochem Mol Biol. 2015 Aug;186:1-7. doi: 10.1016/j.cbpb.2015.04.004. Epub 2015 Apr 14. PMID: 25882225.
An S, Zhang Y, Wang T, Luo M, Li C. Molecular characterization of glutaredoxin 2 from Ostrinia furnacalis. Integr Zool. 2013 Apr;8 Suppl 1:30-8. doi: 10.1111/j.1749-4877.2012.00301.x. Epub 2012 Oct 31. PMID: 23621469.
Giordano E, Peluso I, Rendina R, Digilio A, Furia M. The clot gene of Drosophila melanogaster encodes a conserved member of the thioredoxin-like protein superfamily. Mol Genet Genomics. 2003 Feb;268(5):692-7. doi: 10.1007/s00438-002-0792-0. Epub 2003 Jan 18. PMID: 12589444.
Response 2:Accepted. Two references, Liu et al. 2021 and An et al. 2013 related closely to our article, were added in the sections of Introduction (lines 51-53) and Results and Discussion (lines 222-226), respectively.
Point 3:I suggest that the Discussion and especially the Conclusion sections should be sufficiently expanded. Authors could describe and discuss the effect of camptothecin on Spodoptera cells. The information could be found in the following articles:
Shu B, Yang X, Dai J, Yu H, Yu J, Li X, Cao L, Lin J. Effects of camptothecin on histological structures and gene expression profiles of fat bodies in Spodoptera frugiperda. Ecotoxicol Environ Saf. 2021 Nov 8;228:112968. doi: 10.1016/j.ecoenv.2021.112968. Epub ahead of print. PMID: 34763196
Shu B, Zou Y, Yu H, Zhang W, Li X, Cao L, Lin J. Growth inhibition of Spodoptera frugiperda larvae by camptothecin correlates with alteration of the structures and gene expression profiles of the midgut. BMC Genomics. 2021 May 26;22(1):391. doi: 10.1186/s12864-021-07726-8. PMID: 34039281; PMCID: PMC8157707.
Yang F, Wang L, Zhang L, Zhang Y, Mao L, Jiang H. Synthesis and biological activities of two camptothecin derivatives against Spodoptera exigua. Sci Rep. 2019 Dec 2;9(1):18067. doi: 10.1038/s41598-019-54596-y. PMID: 31792297; PMCID: PMC6889156.
Moreover, other the role of other antioxidative systems such as antioxidants, peroxidases, and thioredoxins could be discussed.
Response 3:Accepted. Two references, Shu et al. (Ecotoxicol. Environ. Saf. 2021, 228:112968 and BMC Genomics. 2021, 22:391.), related to our article were added in the sections of Discussion (lines 214-221). Addtionally, we focused on the synergisms of CPT and glutaredoxins (Grxs) on the cytotoxicity and insecticide susceptibility of CPT in Spodoptera exigua, therefore the other antioxidative systems such as antioxidants, peroxidases, and thioredoxins was not discussed in this paper.
Point 4:Please cite Primer-BLAST software
Response 4:Accepted.
Point 5: Please add the citation in line 175
Response 5:Accepted.
Point 6:The text requires considerable copy-editing due to misprints, spelling, and grammar errors (in human – in humans (line 33), homoeostasis -> homeostasis (line 24), a basic information -> basic information, etc.) there are too many issues for the reviewer to mention, thus I suggest re-checking the manuscript attentively to fix the errors accordingly.
Response 5:Accepted. We have reviewed and corrected carefully the full manuscirpt.

Round 2
Reviewer 3 Report
The authors have addressed most of the comments mentioned. However, I still find that stating that in SeGrx1-silenced insects, camptothecin causes more intense oxidative stress should be treated as a hypothesis rather than the established fact since the intensity of oxidative stress has not been studied there. The previous research cited showed that camptothecin causes oxidative stress per se, thus it has not been assessed in SeGrx1-silenced insect cells. Therefore, I strongly recommend changing the wording in the Discussion and Conclusion sections by saying that it could be proposed that oxidative stress is more intense in SeGrx1-silenced insects. As I stated earlier, I agree with the fact that this explanation is most probably true, however, since the explicit experiments have not been done, the wording should be changed accordingly.
Author Response
Response to Reviewer 3 Comments
Point 1:The authors have addressed most of the comments mentioned. However, I still find that stating that in SeGrx1-silenced insects, camptothecin causes more intense oxidative stress should be treated as a hypothesis rather than the established fact since the intensity of oxidative stress has not been studied there. The previous research cited showed that camptothecin causes oxidative stress per se, thus it has not been assessed in SeGrx1-silenced insect cells. Therefore, I strongly recommend changing the wording in the Discussion and Conclusion sections by saying that it could be proposed that oxidative stress is more intense in SeGrx1-silenced insects. As I stated earlier, I agree with the fact that this explanation is most probably true, however, since the explicit experiments have not been done, the wording should be changed accordingly.
Response 1: Accepted. “In our previous studies, a significant increase in the level of intracellular ROS has been observed, and which was accompanied by markedly increased DNA damage, lipid peroxidation and protein carbonylation after exposing to CPT in IOZCAS-Spex-II cells [16]. It could be proposed that oxidative stress is more intense in SeGrx1-silenced IOZCAS-Spex-II cells” was added in the section of 3.1 (lines 174-178). And “It could be proposed that oxidative stress is more intense in insects” was added in the section of Conclusions (lines 276-277).
